# Epitaxially grown silicon-based single-atom catalyst for visible-light-driven syngas production

Huai Chen[1,8], Yangyang Xiong[1,8], Jun Li[2,3], Jehad Abed [2], Da Wang[4,5], Adrián Pedrazo-Tardajos[4,5], Yueping Cao[1], Yiting Zhang[1], Ying Wang [6], Mohsen Shakouri [7], Qunfeng Xiao[7], Yongfeng Hu [7], Sara Bals [4,5], Edward H. Sargent [2], Cheng-Yong Su[1] ✉ & Zhenyu Yang [1] ✉

Improving the dispersion of active sites simultaneous with the efficient harvest of photons is a key priority for photocatalysis. Crystalline silicon is abundant on Earth and has a suitable bandgap. However, silicon-based photocatalysts combined with metal elements has proved challenging due to silicon's rigid crystal structure and high formation energy. Here we report a solid-state chemistry that produces crystalline silicon with well-dispersed Co atoms. Isolated Co sites in silicon are obtained through the in-situ formation of $CoSi_2$ intermediate nanodomains that function as seeds, leading to the production of Co-incorporating silicon nanocrystals at the $CoSi_2$/Si epitaxial interface. As a result, cobalt-on-silicon single-atom catalysts achieve an external quantum efficiency of 10% for $CO_2$-to-syngas conversion, with CO and $H_2$ yields of 4.7 mol $g_{(Co)}^{-1}$ and 4.4 mol $g_{(Co)}^{-1}$, respectively. Moreover, the $H_2$/CO ratio is tunable between 0.8 and 2. This photocatalyst also achieves a corresponding turnover number of $2 \times 10^4$ for visible-light-driven $CO_2$ reduction over 6 h, which is over ten times higher than previously reported single-atom photocatalysts.

The optimization of active site performance and atom efficiency are essential objectives in the realms of energy conversion, environmental remediation, and chemical synthesis[1–3]. Single-atom catalysts (SACs) have received worldwide research interests toward these goals due to their tunable and well-defined coordination environments[4–6].

Notable synthetic progresses have been made to achieve SACs with highly dispersed guest atoms[7–9]. The precise control of the desired catalysts configuration, however, remains challenging for developing efficient SACs. Specifically, seeking approaches to achieve efficient atomic metal dispersion on the desired substrate is essential to fulfil these desired objectives in practical catalytic applications.

Constructing SACs on semiconductor substrates has the potential to unite efficient photon harvesting and electron–hole separation for catalytic reactions[10–14]. To date, photocatalytic SACs have featured metal oxides, chalcogenides, and metal–organic frameworks;[15–19] however, poor charge carrier generation has led to external quantum efficiencies below 1%, even with the benefit of

[1]MOE Laboratory of Bioinorganic and Synthetic Chemistry, Lehn Institute of Functional Materials, School of Chemistry, Sun Yat-sen University, 510275 Guangzhou, China. [2]Department of Electrical and Computer Engineering, University of Toronto, 35 St. George Street, Toronto, ON M5S 1A4, Canada. [3]Frontiers Science Center for Transformative Molecules, Shanghai Jiao Tong University, 200240 Shanghai, China. [4]Electron Microscopy for Materials Science (EMAT), University of Antwerp, Groenenborgerlaan 171, 2020 Antwerp, Belgium. [5]NANOlab Center of Excellence, University of Antwerp, 2020 Antwerp, Belgium. [6]Department of Chemistry, Chinese University of Hong Kong, Shatin, New Territories, Hong Kong SAR, China. [7]Canadian Light Source, Inc. (CLSI), Saskatoon, Saskatchewan, Canada. [8]These authors contributed equally: Huai Chen, Yangyang Xiong. ✉e-mail: cesscy@mail.sysu.edu.cn; yangzhy63@mail.sysu.edu.cn

notable advances in host/substrate engineering for increased light absorption capability[20–26].

Silicon has a suitable bandgap (1.1 eV), elemental abundance, and high thermal stability for electronic and catalytic applications[27–33]. The rigid diamond-type structure of silicon tends to expel dopant elements during crystal formation at high temperatures, yielding low concentrations of active sites in silicon SACs reported to date[34–36]. We reasoned that epitaxial growth on silicon could provide precise control over the location of the guest atoms[37]. We sought a solid-state chemistry approach to construct silicon-based SACs (Fig. 1), pursuing a route wherein metal silicide with layered structures leads to silicon domains that decompose to yield active metal atoms dispersed in the silicon lattice[38]. We posited that the in situ formation of metal silicide might facilitate the growth of silicon and lead to the incorporation of dopant metals from the metal silicide/silicon interface.

We tested this idea using cobalt silicide ($CoSi_2$) as the target and silicon as the host since these have similar cubic crystalline structures and low lattice mismatch (1.6% and 2.2% along the [100] and [110] orientations, respectively, Fig. 1a)[39]. We hypothesized that the $CoSi_2$ domains could serve as templates to promote the in situ growth of

diamond-structured crystalline Si (c-Si) via atomic epitaxy, enabling the incorporation of Co atoms in c-Si lattice during the decomposition of the $CoSi_2$ domain at high temperatures (Fig. 1b). We prepared Co@Si SACs with tunable Co concentrations in c-Si (0.4–1.4 wt%) with negligible aggregation (e.g., formation of clusters or nanocrystals, Supplementary Table 1). The incorporation of atomically dispersed Co on nano-silicon enables efficient visible-light-driven $CO_2RR$ to produce syngas ($H_2$ + CO) with the $H_2$: CO ratio ranging between 0.8 and 2 ($v/v$). With 1.4 wt% of Co loading, the Co@Si catalysis achieved turnover number (TON) values up to $2 \times 10^4$, over 13 times higher than the previous best obtained using single-atom photocatalysts[40].

## Results and discussion
### Preparation of Co@Si SACs
We firstly prepared an amorphous precursor $HSiO_{1.5}$ containing homogeneously dispersed cobalt ions by a sol–gel reaction (see "Methods" and Supplementary Fig. 1). The precursor was annealed at 500 °C under a slightly reducing atmosphere (5% $H_2$ + 95% Ar) to form $CoSi_2$ nanodomains embedded in a porous silica ($SiO_2$) matrix. The temperature was subsequently increased to 700 °C to enable the

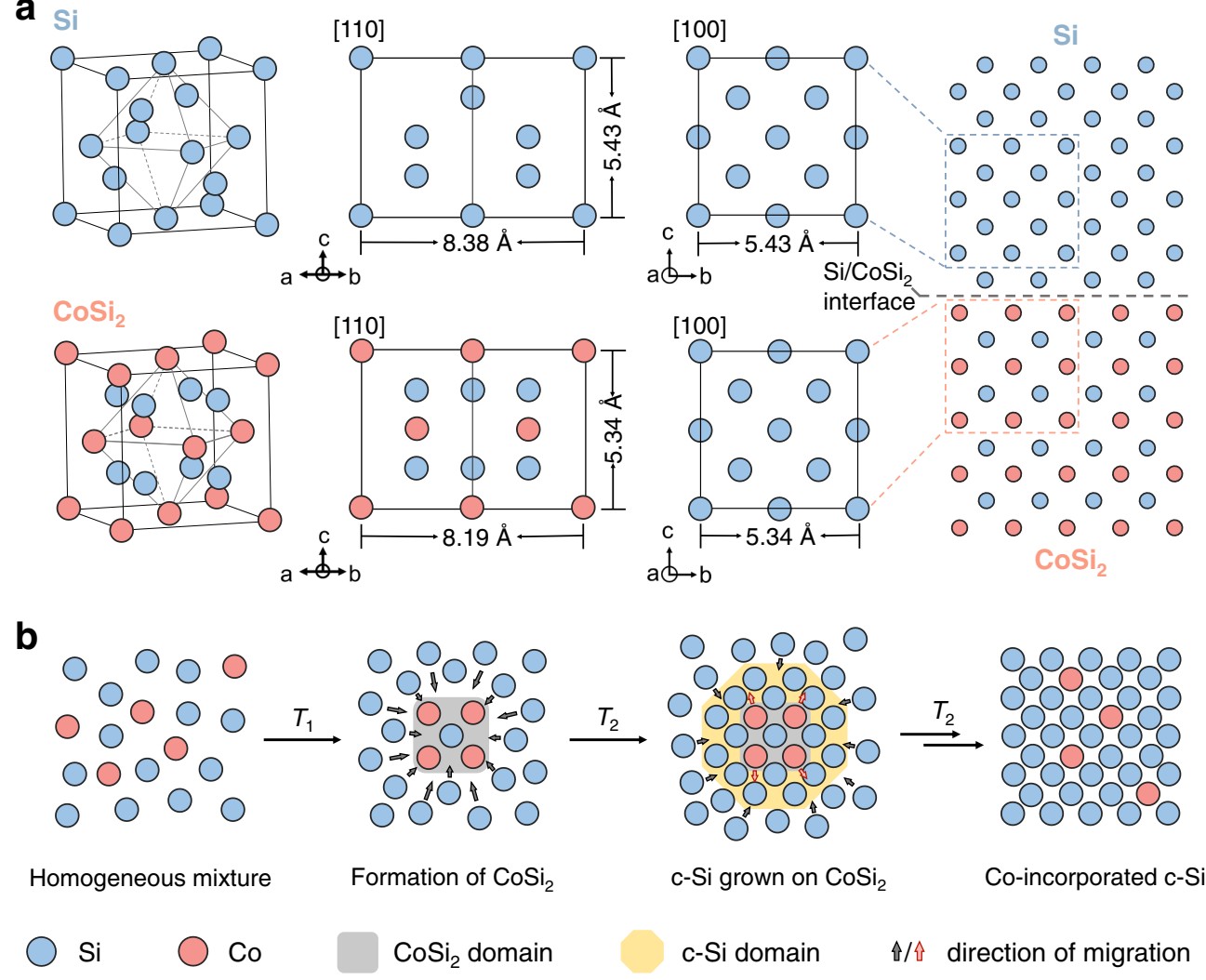

**Fig. 1 | Design of silicon-based SAC formed by solid-state epitaxy. a** Crystal models of c-Si and $CoSi_2$ and the proposed epitaxial interface between c-Si (100) and $CoSi_2$ (100) facets. **b** Proposed synthetic steps of Co@Si SACs: (1) formation of $CoSi_2$ intermediate via thermal annealing of the amorphous sol–gel precursor containing homogeneously mixed Co and Si atoms; (2) Si atom diffusion and

epitaxial growth on c-Si domains at a predesigned annealing temperature $T_1$; and (3) decomposition of $CoSi_2$ during extended annealing at elevated temperature $T_2$ ($T_2 > T_1$), yielding c-Si with atomic-dispersed Co sites. The surrounding amorphous $SiO_2$ matrix is omitted for clarity.

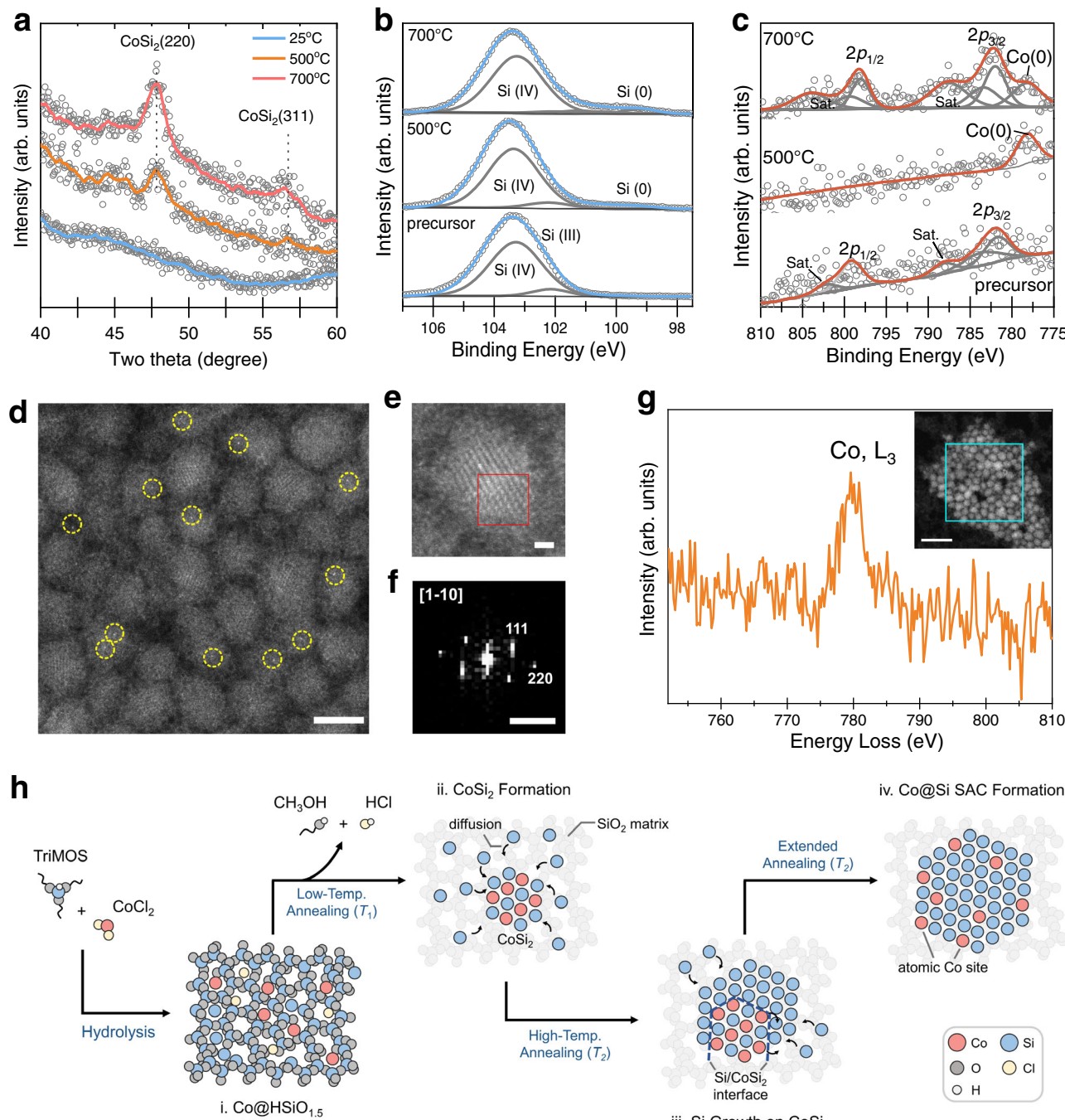

**Fig. 2 | Materials evolution and characterization of Co@Si SACs (Co concentration: 1.4 wt%). a** XRD results of the sol–gel precursor and products after thermal annealing. **b, c** High-resolution XPS results of silicon $2p$ and cobalt $2p$ regions. Spectra of (**b**) to which we have curve-fit are shown for the Si $2p_{3/2}$ emissions. Silicon $2p_{1/2}$ components are omitted for clarity. "Sat." refers to the satellite peak of the cobalt $2p$ signal. **d** HAADF-STEM images of Co@Si NCs extracted from the SAC solid and functionalized by octene. Co atoms are highlighted by yellow circles (scale bar: 5 nm). **e** A high-resolution HAADF-STEM image of a Co@Si NC (scale bar, 1 nm) and **f**, a fast Fourier transform (FFT) pattern from the area denoted by a red square in (**e**) (scale bar: 5 1/nm). **g** Electron energy loss spectrum (EELS) result reveals the presence of cobalt (L₃, 779 eV) from a selected area of a HAADF-STEM image of Co@Si NCs as inset (scale bar, 20 nm). **h** Schematic illustration of the CoSi₂-assisted Co@Si SAC formation mechanism. The process is initiated by CoSi₂ nanodomains formation by doping Co(II) in a silicon-rich network homogeneously (Step i) and annealing at low temperature ($T_1$ = 500 °C) (Step ii); the thermally-induced CoSi₂ decomposition occurred at high temperature ($T_2$ = 700 °C) during the silicon crystallization enables the crystal re-arrangement (Step iii); the diffusion of Co and Si atoms during extended annealing ($T_2$ = 700 °C) enables single Co dispersion in the lattice of c-Si, yielding Co@Si SAC (Step iv).

migration and crystallization of Si atoms on CoSi₂. Due to the difference in thermal stability between c-Si and CoSi₂ crystals[41], CoSi₂ domains slowly decompose and release Si and Co atoms following extended annealing at 700 °C. The dissociated Co atoms diffused and were finally incorporated into the c-Si lattice, yielding Co@Si nanodomains in the SiO₂ matrix.

We monitored the formation of Co@Si structures using powder X-ray diffraction (XRD, Fig. 2a). No crystalline features are seen in the XRD pattern of the HSiO₁.₅ precursor, confirming its amorphous nature. After the thermal annealing at 500 °C, strong signals emerge at 47.8° and 56.2°, which can be assigned to the (220) and (311) facets of CoSi₂[42] (Fig. 2a). The signal intensities of these two facets increase after

subsequent annealing at 700 °C, suggesting the growth of CoSi$_2$ nanodomains at elevated temperatures. However, the presence of c-Si domains cannot be determined by XRD measurements due to the high degree of lattice coherence between CoSi$_2$ and c-Si. We, therefore, proceeded with hydrofluoric acid (HF) etching to remove CoSi$_2$ and SiO$_2$ matrix and confirmed the formation of silicon nanocrystals (Si NCs) with an average size of ~12 nm as verified by the Scherrer equation (Supplementary Fig. 2). The migration of Co atoms becomes more evident at higher temperatures as the crystalline cobalt is found from the XRD pattern of the sample annealed at 800 °C (Supplementary Fig. 3).

The XRD results hint that both CoSi$_2$ and c-Si can be generated and co-exist in a series of solid-state reactions. We sought to explore further—with the aid of using X-ray photoelectron spectroscopy (XPS)—atomic migration and the local environment of both Si and Co during the annealing processes. Prior to annealing, only the emission features at 102.1 and 103.2 eV were observed, which can be attributed to Si(III) and Si(IV) (Fig. 2b). The characteristic signals of Co(II) are found in higher-energy regions (2$p_{1/2}$: 795 eV; 2$p_{3/2}$: 780 eV, Fig. 2c), consistent with the homogeneously dispersed ionic cobalt in the HSiO$_{1.5}$ precursor[42]. After thermal annealing at 500 °C, the original Co(II) signals disappeared, whereas a new feature is found at 778.2 eV, which can be assigned to Co(0) (Fig. 2c). A new signal at 99.2 eV emerges on the Si 2$p$ spectrum (Fig. 2b), falling within the literature values of Si(0) in c-Si[42]. Since cobalt and silicon have similar electronegativity ($\chi_{Co}$ = 1.88, $\chi_{Si}$ = 1.90) meanwhile the crystalline cobalt signal is absent from the XRD pattern (Fig. 2a), we therefore conclude that CoSi$_2$ nanodomains are formed through a two-step process: (1) the disproportionation of HSiO$_{1.5}$ and (2) the redox reaction from Co$^{2+}$ to CoSi$_2$ (in which the valence of Co is close to 0) at 500 °C due to its relatively lower formation energy compared to that of metallic Co[43]. When the temperature increases to 700 °C, no observable change in the signals of Si(0) and Si(IV), whereas the disappearance of the Si(III) feature suggests complete disproportionation of HSiO$_{1.5}$ to elemental Si and SiO$_2$ at that temperature[43]. As for Co, the notable Co(0) signal confirm the presence of elemental Co sites embedded in the c-Si lattice, whereas the re-emergence of Co(II) signals at 795 eV (2$p_{1/2}$) and 780 eV (2$p_{3/2}$) indicates partial dissociation of atomic Co from CoSi$_2$ and the subsequent diffusion to the SiO$_2$ matrix, yielding CoO$_x$.

We next investigated the structure of Co@Si SACs and the dispersion of Co atoms using electron microscopy. The distribution of Co within the annealed sample is relatively uniform as indicated by the combination of scanning electron microscopy (SEM) and the corresponding EDX analysis (Supplementary Figs. 4 and 5 and Supplementary Table 2). To further test whether the active Co sites are located on the silicon nanodomains, we applied HF-etching to extract the Si NC components; and we then functionalized their surfaces with alkyl ligands (see "Methods"). We investigated the surface-functionalized NCs using high-angle annular dark-field scanning transmission electron microscopy (HAADF-STEM) imaging, showing highly dispersed Co atoms (as denoted with yellow dashed circles) in Si NCs with an average diameter of 5 nm (left panel of Fig. 2e and Supplementary Fig. 6). A Co@Si NC exhibits lattice spacings of ~3.1 Å and ~2.0 Å (Fig. 2e, f). The first distance is in agreement with the expected d-spacing of the {111} lattice planes (3.1 Å) of c-Si. However, the value of ~2.0 Å is slightly different from the expected d-spacing of the {220} lattice planes (1.9 Å) of c-Si, a lattice expansion which has already been reported for Si NCs in a previous study[44]. The Co electron energy loss spectroscopy (EELS) signal (L$_3$ edge, 779 eV) is observed in the EEL spectrum from a selected area of functionalized Co@Si NCs (demonstrated by the cyan square), which proved the presence of Co (Fig. 2g and Supplementary Note 1). By tuning the Co:Si ratio in the sol–gel precursor, we tuned the Co concentration from 0.4 to 3.4 wt% as evidenced via inductively coupled plasma optical emission spectroscopy (see "Methods" and Supplementary Table 1).

The materials characterization leads us to a more detailed model of the CoSi$_2$-assisted SAC growth mechanism (Fig. 2h). During the thermal annealing at 500 °C, HSiO$_{1.5}$ disproportionates and form Si atoms, which subsequently interact with Co(II) under a reducing atmosphere to form CoSi$_2$ nanodomains within the porous SiO$_2$ matrix. Owing to the low lattice mismatch with c-Si, CoSi$_2$ domains can behave as the "seeds" to allow further nucleation of Si atoms at elevated temperatures (e.g., 700 °C), yielding Si NCs on grown from the CoSi$_2$ nanodomains. In the meantime, because of the lack of thermal stability of CoSi$_2$ at 700 °C, Co and Si atoms gradually diffuse from the CoSi$_2$ lattice and participate in the growth of Si NCs to form Co@Si SACs.

## Structural analysis

We further investigated the local atomic environment of Si and Co components in Co@Si SACs using synchrotron-based techniques. The X-ray absorption near-edge structure (XANES) spectra at the Si K-edge exhibit both elemental Si and SiO$_2$ features from the Co@Si structures with various Co concentrations[45] (Supplementary Fig. 7). The Co K-edge XANES spectra of Co@Si samples show undistinguishable cobalt oxides (CoO$_x$) and elemental Co signals, suggesting that the Co atoms are well-dispersed in SACs. In contrast, notable Co–O and Co–Co features are found from the Co@SiO$_2$ control sample (i.e., no c-Si domains), indicating the inevitable oxidation or aggregation of Co atoms when no crystalline silicon domain is available for anchoring (Fig. 3a).

To investigate the chemical environments of the active sites in Co@Si, we applied Fourier-transformed EXAFS (FT-EXAFS) spectroscopy. We examined the local bonding information of Co atoms (Fig. 3b, Supplementary Figs. 8 and 9, and Supplementary Table 3), finding that the oxidized ($d_{Co-O}$ = 1.6–1.9 Å) and metallic Co ($d_{Co-Co}$ = 2.2–2.5 Å) signals are observed from the spectra of the Co@Si structures with 1.4 wt%Co and 3.4 wt%Co, respectively[45–47]. Both oxidation and aggregation of Co are exhibited in Co@SiO$_2$ control sample. In contrast, neither the metallic feature nor the oxidation state is found in the Co@Si samples with lower Co concentrations (e.g., 0.4 wt% and 0.5 wt%Co). We therefore conclude that the efficient isolated Co dispersion can be achieved only when the Co concentration is below 1.4 wt% (Fig. 3c).

The coordination information of Co atoms can be further extracted from the Co K-edge EXAFS results. Relatively low coordination number (CN) values (2.5–4.0) of Co atoms are noticed in the Co@Si samples with various Co concentrations (from 0.4 wt% to 3.4 wt% Co, Fig. 3d) compared to metallic Co (CN = 6). The emergence of the Co–O was noticed in the 1.4 wt% sample and both Co–O and Co–Co appeared in the 3.4 wt% sample, indicating that the increase of Co concentration may introduce oxidation and aggregation of Co atoms (Fig. 3d). The corresponding fitting parameters of bond lengths are summarized in Fig. 3d. The Co–Si bond length within the control Co@SiO$_2$ sample (2.9 Å) is ~25% longer than the average Co–Si bond length of Co@Si SACs (~2.3 Å), which can be attributed to the Co–O–Si linkage between Co and Si atoms (Fig. 3e). In contrast, the average Co–Si bond lengths are consistent in all Co@Si samples, indicating that the Co atoms in Co@Si samples, regardless of the Co concentration, own a similar coordination environment when incorporated in silicon lattice (Fig. 3e).

## Influence of Co concentration on catalytic performance

We investigated the visible-light-driven photocatalytic performance of the Co@Si samples. The porous nature of the SiO$_2$ matrix (pore size: ~5 nm, Supplementary Figs. 10–12) enables high CO$_2$ capture capability (~9.2 cc/g, Supplementary Fig. 13), facilitating CO$_2$RR on Co@Si SACs. In a typical reaction, 2 mg of the Co@Si powders were dispersed in acetonitrile in a sealed glass vial filled with CO$_2$ gas. Triethanolamine (TEOA) and tris(2,2′-bipyridine) ruthenium dichloride ([Ru(bpy)$_3$]Cl$_2$)

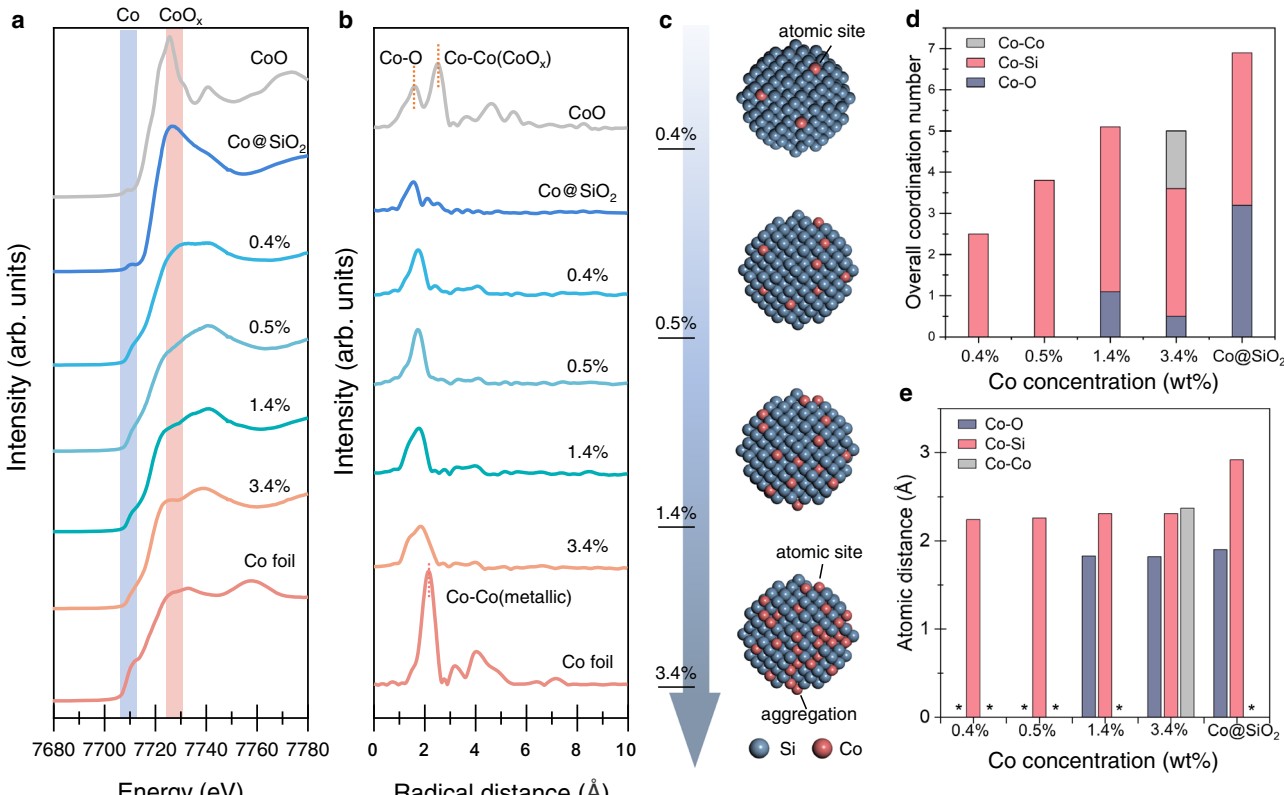

**Fig. 3 | X-ray absorption analysis of Co@Si SACs. a** Co K-edge XANES spectra and **b** Fitted Co K-edge EXAFS spectra of various Co@Si and control samples. The original spectra are available in Supplementary Fig. 8. The shaded rectangles are denoted spectral feature of metallic Co and $CoO_x$, respectively. **c** The schematic shows the influence of Co concentration on the distribution of Co atoms on Si NCs.

**d** Simulated coordination number and **e**, corresponding atomic distance values extracted from the Co K-edge EXAFS spectra of Co@Si and the control Co@SiO$_2$ samples. Negligible values of Co–O and Co–Co contribution in the bond distance are denoted using asterisks (*).

were applied as the sacrificial agent and photosensitizer, respectively. The reduction products, CO and $H_2$, were immediately formed when the solution was irradiated by the white light source ($\lambda_{ex} > 400$ nm). The gas chromatography-mass spectrometric (GC–MS) results confirmed that the generation of $^{13}CO$ when $^{13}CO_2$ was applied as the input gas, ensuring that $CO_2$ is the only carbon source of CO (see "Methods" and Supplementary Fig. 14). The $H_2$ product, in contrast, can be generated even by the Co-free control sample (Supplementary Fig. 15), suggesting that the c-Si domains promote the proton-to-$H_2$ conversion.

The intense light absorption by the c-Si domain further promotes the $CO_2RR$ (Supplementary Fig. 16). The highest yields of CO (4.72 mol g$_{(Co)}^{-1}$) and $H_2$ (4.43 mol g$_{(Co)}^{-1}$) were achieved with the 0.5 wt %Co sample, over $10^3$–$10^4$ times higher than those produced by the commercial Co and $Co_3O_4$, respectively (Fig. 4a). It is important to note that the free-standing Co@Si NCs (i.e., without the $SiO_2$ matrix) also exhibited evident photocatalytic activity compared to the Co-free counterpart when no photosensitizer was applied, indicating that the photocatalytic behaviors of the Co@Si powders originate from the Co-incorporated Si cores (Supplementary Fig. 17, also see Supplementary Note 2 for more details).

The combination of efficient photon harvesting and dispersion of active Co atoms steers the state-of-the-art $CO_2$ conversion. A total TON of ~$2.0 \times 10^4$ for $CO_2$-to-CO was achieved on the Co@Si SACs with 1.4 wt%Co for a typical reaction of 6 h. This is 13-fold higher than the best previously reported heterogeneous catalysts for photocatalytic $CO_2RR$ (Fig. 4b, Supplementary Fig. 18, and Supplementary Table 4)[40]. The $H_2$/CO ratio can be tuned from 0.8 to 2 through the change of Co concentration in Co@Si powders (Fig. 4b) and the amount of proton source (Supplementary Fig. 19, the detailed discussion about the

functions of water and TEOA is available in Supplementary Note 3). We observed a notable decrease in TON for the reaction with higher Co loading (3.4 wt%Co, Fig. 4b). This may be attributed to the aggregation of Co atoms in which the catalytic activity of the single-atom site is inhibited[48].

The stability tests were performed to evaluate the photocatalytic durability of the catalyst. The SACs retained ~100% syngas production rate after three cycles (1 h per cycle) of photocatalysis (Supplementary Fig. 20). The Co@Si sample with 3.4 wt%Co presents the highest CO conversion of ~10 mmol g$^{-1}$, which can be attributed to its relatively high Co loading (Fig. 4c). The atomic Co dispersion is remained consistent in catalyst after irradiation (Supplementary Fig. 21). The CO production rate drastically reduced after 6 h irradiation (Fig. 4c). This may be because of the decomposition of photosensitizer under continuous visible-light irradiation[49]. Through the compositional engineering of the Co@Si SACs, we achieved an external quantum efficiency (EQE) of 10% combining the efficient metal loading, which surpasses the performance of previously reported heterogeneous photocatalysts for visible-light-driven $CO_2RR$ (Fig. 4d, more information about the referred catalysis is available in Supplementary Table 5)[15–18,20,50–64].

The materials characterization and catalytic performance investigation presented above lead us to a more detailed model of the syngas production promoted by Co@Si SACs which follows a joint reaction (Supplementary Fig. 22)[65]. As the initial step, $CO_2$ gas was captured by the porous structure of the $SiO_2$ matrix and converted to the absorption state ($CO_2^*$, Step i). The proton was supplied by the ionic equilibrium of $H_2O$. Under the visible-light irradiation, the photosensitizer $[Ru(bpy)_3]^{2+}$ is promoted to its excited state form ($[Ru(bpy)_3]^{2+*}$, Step ii). The photoexcited electron is subsequently

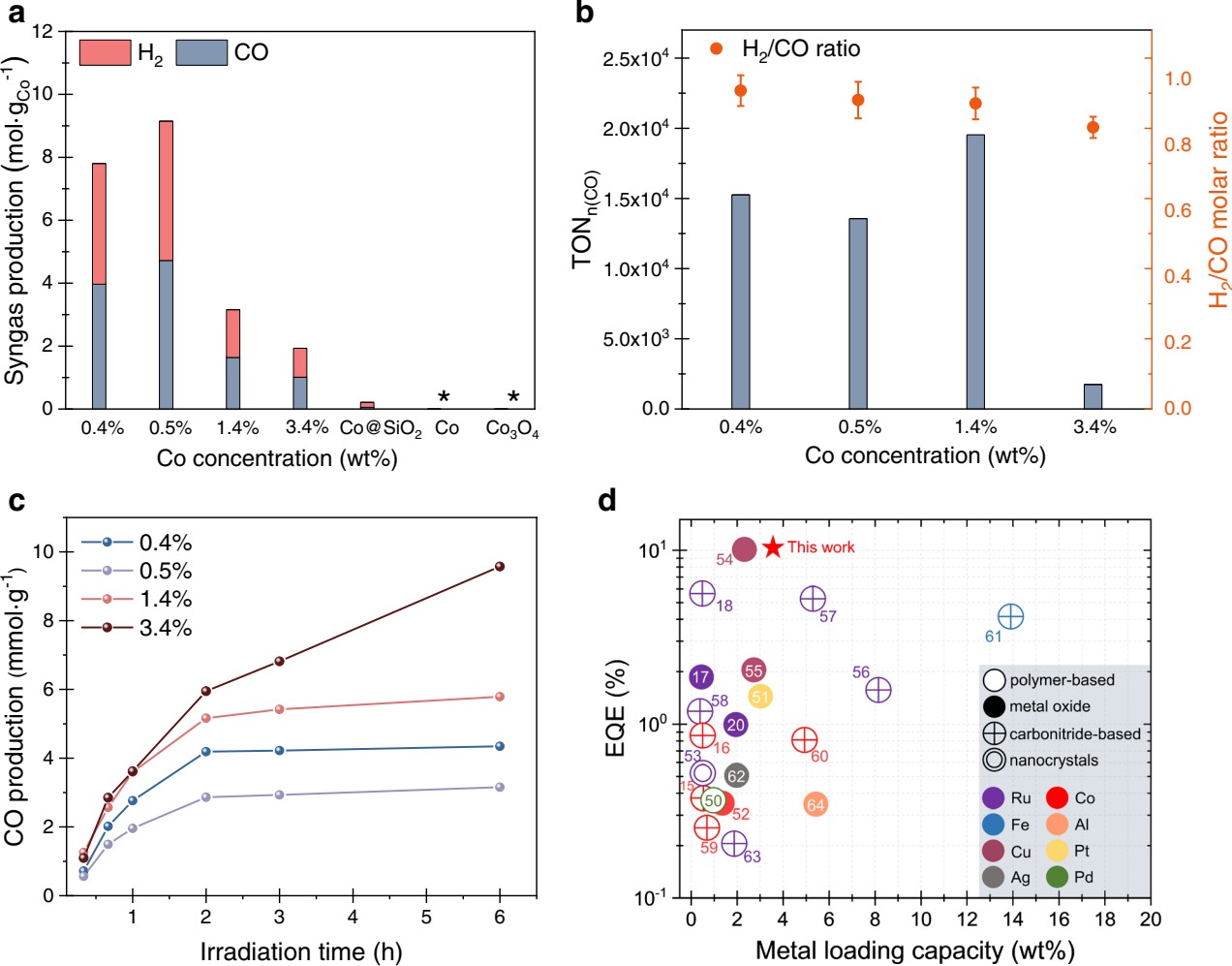

**Fig. 4 | Performance of photocatalytic syngas production. a** Syngas production rates of Co@Si SACs with various Co concentrations. Negligible values of product yield are denoted using asterisks (*). **b** TON results and production ratio on CO production with various Co concentrations. The error is due to the deviation of multiple measurements. **c** Time-dependent CO production performance of Co@Si samples with various Co concentrations. **d** The comparison of the EQE values and metal loading capacity between the Co@Si SACs and the reported photocatalysts[15–18, 20, 50–64]. The different symbols and coulors are denoted as types of catalysts and loading metal, respectively, which display in the darker region. Detailed parameters of the materials are available in Supplementary Table 5.

transferred to Co@Si domains, and the hole is quenched by the TEOA to regenerate $[Ru(bpy)_3]^{2+}$ (Steps iii–iv). Meanwhile, the $CO_2^*$ grasps the electron and the proton from Co@Si and solution, respectively, which subsequently converts to form the COOH* intermediate and finally forms CO (Steps v and vi)[52,66]. A considerable amount of as-formed protons are eventually reduced on the Co-free c-Si facets to produce $H_2$ (Step vii). Similar joint reaction processes have also been found from other photocatalytic systems such as Janus particles, and two-dimensional heterostructures[67–71].

We report a solid-state synthetic strategy to prepare Si-based SACs with tunable Co concentrations between 0.4 wt% and 3.4 wt% and a porous $SiO_2$ matrix as catalyst support. The formation of Co@Si powders is based on a two-step solid-state epitaxial growth process. The in situ formation of the $CoSi_2$ intermediate is key to the formation of Si domains at relatively low temperatures and the subsequent Co diffusion into the silicon lattice. Combined spectroscopic and microscopic techniques were applied to elucidate the atomic environment of the active Co sites on silicon nanodomains. The catalytic efficiency towards visible-light-driven syngas production is exhibited by 4.72 mol $g_{(Co)}^{-1}$ of CO and 4.43 mol $g_{(Co)}^{-1}$ of $H_2$ output with a TON value of $2 \times 10^4$, outclassing any other photocatalytic SACs based on noble metals or transition metals. The $H_2$/ CO ratio can be continuously tuned between 0.8 and 2 by adjusting the Co concentration and the components of the proton source, which are compatible with downstream industrial catalytic processes such as Fischer–Tropsch synthesis and methanol production[72,73]. This work develops a synthetic path toward efficient SACs using group IV semiconductors, which are believed to be ideal photocatalytic platforms for $CO_2$ valorization integrated with organic synthesis.

## Methods

### Chemicals

All chemicals used are commercially available and were used without any additional purification steps. Trimethoxysilane (TriMOS, 95%), tetramethoxysilane (TMOS), cobalt chloride hexahydrate ($CoCl_2 \cdot 6H_2O$, 98%), cobalt (97%), cobalt oxide (98%), electronic-grade hydrofluoric acid (HF, 49%), azazobisisobutyronitrile (AIBN, 98%), triethanolamine (TEOA, 98%), tris(2,2′-bipyridine) ruthenium dichloride ($[Ru(bpy)_3]^{2+}Cl_2$, 99%), 1-octene (98%), 1-dodecene (95%), toluene (99%), dichloromethane (99.9%), and acetonitrile (95%) were purchased from Aladdin Inc. The activated carbon (granular size: ~1.5 mm, bulk density: 150−440 kg/m³) was purchased from Merck KGaA, Germany.

## Solid-state synthesis of Co@Si SACs

Overall, 63 mmol of TriMOS (7.7 g) was weighed out in a nitrogen-filled glovebox and transferred immediately into a 100-mL Schlenk flask equipped with magnetic stirring under a nitrogen atmosphere. The flask was then soaked in an ice bath to reduce the temperature to ~0 °C. 0.4 mmol of $CoCl_2\cdot6H_2O$ (100 mg), 10 mL of dilute nitric acid (0.9 mmol), and 10 mL of methanol (25 mmol) were then added into the flask and mixed under a continuous flow of nitrogen. Clear and light-pink gels were immediately formed within 5 min. The temperature was then increased to room temperature, and the product was set still without mechanical stirring under a nitrogen atmosphere for 24 h. After the aging process, the product was first isolated from the residual liquid by vacuum filtration and subsequently transferred to a vacuum oven and dried for 16 h. The pink, gel-like product was gradually transferred to blue powders. In all, 2 g of the powders were then placed in a quartz reaction boat and moved to a high-temperature tube furnace (Lindberg). The sample was heated from ambient to the predesigned peak temperature (e.g., 700 °C) at 18 °C/min in a slightly reducing atmosphere (5% $H_2$ + 95% Ar). The sample was maintained at the processing temperature for a predesigned period and then naturally cooled down to room temperature. The resulting black-brown powdery product was ground to fine powders and stored in a 20-mL vial in ambient condition for further use. The control sample Si/SiO$_2$ is prepared following the same conditions without adding Co species.

## Liberation of Co@Si NCs

An optimized HF-etching protocol was applied to liberate Co@Si NCs from the $SiO_2$ matrix[74]. Overall, ~0.5 g of the ground product was transferred to a polyethylene terephthalate beaker equipped with a Teflon-coated stir bar. In total, 3 mL of ethanol and 3 mL of deionized water were added to the beaker with mechanical stirring to form a brown suspension. In all, 3 mL of 49−51% HF aqueous solution was subsequently added into the mixture in ambient conditions under mechanical stirring to initiate the etching reaction (Caution! HF solution must be handled with extreme care). After 1 h, the color of the suspension gradually changed to orange. Next, ~60 mL of chloroform (74 mmol) was added to extract the hydride-terminated Co@Si particles from the bottom of the aqueous layer. The product/chloroform solution was obtained using a plastic pipette by multiple (i.e., 3*20 mL) extractions.

## Passivation of Co@Si NCs

To passivate the hydride-terminated Co@Si NCs, the NCs/dichloromethane suspension was subsequently transferred to glass test tubes. The product was isolated by centrifugation at 4000 rpm for 10 min. After the centrifugation, the dichloromethane supernatant was decanted. The solid was redispersed in ~15 mL of toluene and was then transferred to a dried 100-mL Schlenk flask equipped with a Teflon-coated magnetic stir and attached to a nitrogen-charged Schlenk line. Under constant nitrogen flow, 1 mg of the radical initiator azobisisobutyronitrile (AIBN, 6 μmol) and 5 mL of the ligand 1-octene (3.2 mmol) were added to the flask under mechanical stirring. The temperature was then increased by 65 °C in an oil bath under a static nitrogen atmosphere. The hydrosilylation reaction was maintained for a minimum of 12 h to yield a transparent orange solution. After the reaction, the flask was cooled down to room temperature. Overall, 10 mL of the alkyl-functionalized Co@Si NCs solution was equally dispensed into two 50-mL centrifuge tubes. In all, 40 mL of methanol/ethanol-mixed solvent (1:1 volume ratio) was added to each centrifuge tube, yielding ~50 mL of cloudy light-yellow dispersion. The tubes were then centrifuged at 7800 rpm for 10 min. The supernatant was decanted, and the precipitate was redispersed in a minimal amount of toluene (~2 mL) and reprecipitated upon the addition of a mixed antisolvent combination containing 1:1 (v/v) of methanol:ethanol. The redispersion/centrifugation/supernatant-removal procedure was repeated twice.

Finally, the precipitate was redispersed in hydrophobic organic solvents (e.g., toluene, hexene) for further use.

## Solid-state synthesis of Co@SiO$_2$ control sample

Overall, 63 mmol of TMOS (9.6 g) was weighed out in a nitrogen-filled glovebox and transferred immediately into a 100-mL Schlenk flask equipped with magnetic stirring under a nitrogen atmosphere. The flask was then soaked in an ice bath to reduce the temperature to ~0 °C. 0.4 mmol of $CoCl_2\cdot6H_2O$ (100 mg), 10 mL of dilute nitric acid (0.9 mmol), and 10 mL of methanol (25 mmol) were then added into the flask and mixed under a continuous flow of nitrogen. Clear and light-pink gels were immediately formed within 5 min. The temperature was then increased to room temperature, and the product was set still without mechanical stirring under a nitrogen atmosphere for 24 h. After the aging process, the product was first isolated from the residual liquid by vacuum filtration and subsequently transferred to a vacuum oven and dried for 16 h. The pink, gel-like product was gradually transferred to blue powders. In total, 2 g of the powders were then placed in a quartz reaction boat and moved to a high-temperature tube furnace (Lindberg). The sample was heated from ambient to the predesigned peak temperature (e.g., 700 °C) at 18 °C/min in a slightly reducing atmosphere (5% $H_2$ + 95% Ar). The sample was maintained at the processing temperature for a predesigned period and then cooled down to room temperature naturally. The resulting gray powdery product was ground to form fine powders and stored in a 20-mL vial in ambient condition for further use. Due to the absence of $Si^{3+}$ in the TMOS molecule, the thermal annealing process of the sol–gel product only yields amorphous $SiO_2$ with atomically dispersed Co sites (i.e., Co@SiO$_2$).

## Photocatalytic CO$_2$ reduction performance

Syngas production via photocatalytic $CO_2$ reduction was carried out using a 50-mL sealed glass vial with magnetic stirring. Typically, 2 mg of powdery sample was suspended in 5 mL of a mixed solvent of TEOA and MeCN (v/v = 1/4) in the presence of 4 mg of photosensitizer of $[Ru(bpy)_3]^{2+}Cl_2$ and formed a homogeneous light-orange solution. The glass vial was pumped into $CO_2$ (99.999%) and then sealed before placing it onto the holder (5 cm above the light source), and the reaction lasted for a predesigned period at room temperature. The amount of produced $H_2$ and CO was analyzed by an off-line gas chromatograph (FuLi Analytical Instrument Co., Ltd., GC9790 plus) equipped with a thermal conductivity detector and a nitrogen carrier. The light source was a white LED light setup (10 W, $\lambda > 400$ nm, PCX-50B/50 C, Beijing Perfectlight Technology Co., Ltd.). In the long-term stability test in the recycling experiment, TEOA and photosensitizer were directly added to the previous reaction to restart the following cycles.

## Carbon isotope tracer measurements

The isotope-labeling catalytic experiments using $^{13}CO_2$ molecules were performed under the same conditions. Briefly, 2 mg of powdery Co@Si SAC sample was suspended in a 50-mL glass vial with 5 mL of the mixed solvent containing TEOA and MeCN (volume ratio: 1:4) and the presence of 4 mg of photosensitizer of $[Ru(bpy)_3]^{2+}Cl_2$, yielding a homogeneous light-orange solution. $^{13}CO_2$ (99.999%, Sigma) gas was purged three times (15 min for each purge period) to the glass vial to obtain a saturated $^{13}CO_2$ atmosphere. The gaseous products were analyzed using gas chromatography-mass spectrometry (GCMS-QP2010).

## External quantum efficiency (EQE) measurements

The EQE results were estimated by arranging actinometry to measure the photon flux difference through an empty vial and the sample (containing TEOA, MeCN, $H_2O$, $[Ru(bpy)_3]^{2+}Cl_2$, and catalyst) from the bottom to the top of the vial. 2 mg of the powdery 3.4 wt% Co@Si SACs was suspended in a 50-mL glass vial with 5 mL of the mixed solvent

containing TEOA and MeCN (volume ratio: 1:4) and the presence of 4 mg of photosensitizer of tris (2,2′-bipyridine) ruthenium dichloride, to the homogeneous solution 0.5 mL of DI water was added as proton donors. The photocatalytic experiment was proceeded using a blue LED lamp ($\lambda = 450$ nm), and the light intensity was calibrated using an optical power meter (FZ-A, BNU Photoelectric Instruments). The $CO_2RR$ was maintained for 1 h.

The EQE values were calculated using the following Eq. (1):

$$\text{EQE(\%)} = \frac{2n(CO)N_A hc}{t_{irr} I \lambda A} \times 100\% \qquad (1)$$

Here, $n(CO)$ is the number of moles of the product CO, $N_A$ is Avogadro's number, $h$ is Planck's constant, $c$ is the speed of light, $t_{irr}$ is the reaction time, $I$ is the intensity of light, $\lambda$ is the wavelength of the incident light, and $A$ is the cross-sectional area of irradiation. In this study, 18.16 μmol of $H_2$ and 27.92 μmol of CO were produced after the irradiation for 1 h (irradiation wavelength: 450 nm), $N_A = 6.02 \times 10^{23}$ mol$^{-1}$, $h = 6.63 \times 10^{-34}$ J·s, $c = 3 \times 10^8$ m/s, $I = 88$ mW cm$^{-2} = 0.88$ J cm$^{-2}$ s$^{-1}$, $A = 0.785$ cm$^2$. According to the equation, the EQE$_{(H2)}$ and EQE$_{(CO)}$ values were calculated as 4% and 6%, respectively.

### XRD measurements
XRD patterns were collected with a Rigaku Smart Lab diffractometer (Bragg–Brentano geometry, Cu $K\alpha1$ radiation, $\lambda = 1.54056$ Å). The spectra were scanned between 2θ ranges of 10–80° with integration of 350 min.

### XPS measurements and analyses
XPS measurements were carried out using Thermo Scientific KAlpha XPS system with a monochromatic Al $K\alpha$ X-ray source (1486.7 eV, spot size: 400 μm). The electron kinetic energy was measured by an energy analyzer operated in the constant analyzer energy mode at 100 eV pass energy for elemental spectra. Casa XPS software (VAMAS) was used to interpret the high-resolution XP spectra. All spectra were internally calibrated to the C 1$s$ emission (284.8 eV).

### BET and TGA/DSC measurements
The BET measurements with $N_2$ adsorption isotherms (operating temperature: 77 K) and $CO_2$ adsorption isotherms (operating temperature: 273 K) were carried out using a Quanta Chrome Autosorb-iQ$_2$-MP instrument (Anton Paar QuantaTec Inc.). Overall, ~0.2 g of the freshly prepared Co@Si@SiO$_2$ powdery sample was transferred into the gas adsorption tube and degassed by a built-in facility. The sample was then treated for 10 h at 100 °C for activation. The thermogravimetric measurements were carried out using an STA 449 F3 Jupiter instrument under a nitrogen flow with a heating rate of 5 °C min$^{-1}$. In total, ~5 mg of the sample was used for each TGA measurement.

### Absorption measurements
The UV–vis absorption spectra were measured using a Shimadzu spectrometer (UV-3600) by mixing powdery samples with pre-dried barium sulfate.

### $^1$H NMR analyses
In total, ~1 mg of tris (2,2′-bipyridine) ruthenium dichloride was added to an NMR tube and fully dissolved in a mixed solution containing acetonitrile-d$_3$ and deuterium oxide (volume ratio: 4:1). The $^1$H NMR spectra were recorded using a Bruker Advanced 400 NMR spectrometer. Tetramethylsilane (TMS) was applied as the internal standard. The Mestre Nova software (Mestrelab Research, S.L.) was used for data analysis.

### ICP-OES analyses
ICP-OES measurements were performed using a Perkin Elmer spectroscopy (Optima 8300). In all, ~5 mg of the powdery sample was completely dissolved in 5 mL of a mixed solvent containing nitric acid (68%) and hydrofluoric acid (49%) with a volume ratio of 3:2. The solution was then carefully diluted to 100 mL with the addition of deionized water before the characterization.

### Electron microscopy studies
SEM images of the photocatalyst particles were taken with a Hitachi SU8010 scanning electron microscope with an acceleration voltage of 10 kV. Transmission electron microscopy (TEM) images were recorded with a special aberration-corrected transmission electron microscope (JEOL-ARM200F) operated at 200 kV. HAADF-STEM images and EELS results were acquired using an aberration-corrected "cubed" Thermo Fisher Scientific Themis 60–300 microscope operated at an acceleration voltage of 200 kV with a probe current of ~50 pA for HAADF-STEM imaging and ~150 pA for EELS measurement, respectively. Gatan Imaging Filter (GIF) aperture of 2.5 mm was used. The individual EELS image is 25 × 25 pixels, taking a dwell time of 1 s per pixel. The spectrometer dispersion was set to 0.25 eV, and 2048 channels were used. 10 μL of functionalized Co@Si NC dispersion was drop-casted on a homemade graphene grid in order to achieve a better signal-to-noise ratio during the measurements. To alleviate carbon contamination of the samples induced by electron beam illumination, the graphene TEM grid was heated in activated carbon (Charcoal activated for analysis, extra pure, Merck KGaA, Germany) at 300 °C for 8 h and then cooled down to room temperature prior to the electron microscopy study.

### Synchrotron XAS studies
Synchrotron XAS measurements at the Si K- and Co K-edges were carried out at the Soft X-ray Micro Characterization Beamline (SXRMB, $E/\Delta E > 10^4$) of the Canadian Light Source (CLS, Saskatoon, Saskatchewan, Canada). Non-polarized X-rays were used for excitation, and fluorescence X-rays were recorded using a 7-element silicon drift detector. XAS data were processed using Athena software in Demeter (v.0.9.26). A cubic spline function was used to fit the background above the absorption edge for the XAS spectral normalization[75]. The normalized EXAFS function, χ(E), in energy space was then transformed to χ(k), where $k$ is the photoelectron wave vector. To assess the interatomic interaction of Co atoms, χ(k) was multiplied by $k$ to amplify the EXAFS oscillations in the low-$k$ region. Fourier transformation (from $k$ space to R space) of the $k$-weighted χ(k) with a $k$ range of 2.5–11 Å$^{-1}$ for the Co K-edge was applied to differentiate the EXAFS oscillation from different coordination shells. Co K-edge EXAFS data recorded in R space (0.6–3.0 Å) was fitted using Artemis software in Demeter with the FEFF6 program[76], in which structural parameters of Co samples were calculated, including coordination number (CN), bond distance (R), inner potential shift ($\Delta E_0$), and Debye–Waller factor ($\sigma^2$).

## Data availability
All the data supporting the findings of this study are available within the article and its Supplementary Information or from the corresponding authors upon reasonable request. Source data are provided with this paper.

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

## Acknowledgements

This work was supported by the National Natural Science Foundation of China (21821003, 21890380, 21905316), Guangdong Natural Science Foundation (2019A1515011748), the Science and Technology Planning Project of Guangdong Province (2019A050510018), Pearl River Recruitment Program of Talent (2019QN01C108), the EU Infrastructure Project EUSMI (Grant No. E190700310), and Sun Yat-sen University. D.W. acknowledges an Individual Fellowship funded by the Marie-Sklodowska-Curie Actions (MSCA) in Horizon 2020 program (grant 894254 SuprAtom). S.B. and A.P.-T. acknowledge financial support from the European Commission under the Horizon 2020 Programme by grant no. 731019 (EUSMI) and ERC Consolidator grant no. 815128 (REALNANO). This project has received funding from the European Commission Grant (EUSMI E190700310). Synchrotron XAS data described in this paper was performed at the Canadian Light Source, a national research facility of the University of Saskatchewan, which is supported by the Canada Foundation for Innovation (CFI), the Natural Sciences and Engineering Research Council (NSERC), the National Research Council (NRC), the Canadian Institutes of Health Research (CIHR), the Government of Saskatchewan, and the University of Saskatchewan.

## Author contributions

H.C., Y.X., C.S., and Z.Y. designed and directed the study. H.C. led the synthesis of catalysis and analyzed the data. H.C., Y.C., and Y.Z. prepared the SAC samples. Y.X. conducted photocatalytic experiments, data collection, and interpretation. C.S. designed and optimized the protocols for photocatalytic experiments. H.C., Y.X., Y.W., and Z.Y. proposed the photocatalytic mechanism. D.W. and A.P.-T. carried out TEM characterizations and data interpretation under the supervision of S.B. J.L., J.A., M.S., Q.X., Y.H., and E.H.S. carried out the XANES and EXAFS characterizations and data analysis. All authors contributed to and commented on the paper.

## Competing interests

The authors have filed a provisional patent (ZL202011609308.8) for this work to the China National Intellectual Property Administration (CNIPA).
