## [Peer Review File · Nature Communications]

REVIEWER COMMENTS

Reviewer #1 (Remarks to the Author):

This is an exciting contribution from the authors reporting new single-atom catalysts (SACs) for solar CO₂ reduction. In this manuscript, Co@Si SACs were rationally synthesized and thoroughly characterized with a variety of techniques. The SACs demonstrated interesting activity in producing syngas using Ru-bpy as photosensitizer and TEOA as sacrificial electron donor. This work represents a novel preparation of SACs. The manuscript is publishable after addressing a few comments.

1. Figure 3b: please remove the cycles for the synthesized SACs. The cycles only served to blur the spectra. It would really help the readers to present the original spectra here and include the fitting curves in SI.
2. Perhaps the most confusing part of this manuscript is the conditions for photocatalysis. In most cases, it is done in the presence of Ru-bpy and TEOA. However, in Supplementary Figure 14, the amount of H₂O was varied in testing. Was TEOA also used in such testing? Figure 4b could be moved to SI since it concerns testing without the use of Ru-bpy and the activity is much lower than in the presence of Ru-bpy. It would really help the readers to (i) specify conditions in figure captions, and (ii) re-organized some of the results to improve clarity.
3. Figure 4d: please use consistent significant figures for Co concentration.
4. The last paragraph before "Conclusions" and Supplementary Figure 16 are not well supported by experimental results. This mechanism is likely true and could be very useful if the authors could present more results (i.e. modeling).
5. Please remove tracked changes in the manuscript.

Reviewer #2 (Remarks to the Author):

I am only able to comment on the microscopic and spectroscopic evidence presented in the manuscript, as the topic of catalysis falls outside my specific expertise. Thus in the following my remarks apply narrowly to the presented XRD, SEM, TEM, EELS, XPS, and EXAFS characterization of the samples.

Overall, the presented evidence appears to be supportive of the claims, but several shortcomings and unclear points need to be explained.

1. Regarding the XRD results, I do not understand the statement on lines 109-111 that the "crystalline cobalt signal is absent from the XRD pattern" due to the fact that "cobalt and silicon have similar electronegativity". Is this really about electronegativity, and if so, can the authors please elaborate?

2. Regarding the XPS results, the spectral feature assigned to Co(0) seems to be much broader in the 500C sample than in the 700C sample (as shown in Fig. 2c). Is this a real effect, and if so what is its cause, or is the peak deconvolution perhaps influenced by the relatively large noise?

3. Regarding the EELS results, the authors should consider showing also the Si K-edge response for the nanoparticle measured in Fig. 2e. Most importantly, however, it is curious that there is no attempt to capture the Co L-edge response of the individual bright atoms, as their identity has not been firmly established. Surely the authors collected EELS maps over such bright contrast? Even if the signal-to-noise is challenging, it should not be entirely out of the realm of possibility to localize the EEL signal to those atomic positions – and indeed show the full measured spectral range at those locations to rule out the presence of other impurities.

4. Regarding the EXAFS results, two things are unclear:

- How is the "Co-Si" fraction determined exactly?

- How is there so much more Co-Co present in the 1.4% and 3.4% samples in Fig. 3d, even though in the spectral responses shown in Fig. 3a and b, it is very hard to make out such a large qualitative difference. The authors should show the peak deconvolutions underlying these analyses.

5. Regarding methods, some more elaboration is needed:

- Synchrotron XAS studies should mention the measurement mode, any relevant polarization, the expected energy resolution, and any further experimental parameters. Any analysis and fitting parameters should also be explicitly disclosed.

Finally, the authors should openly deposit the data – and ideally the analysis code – underlying their findings at latest upon final revision of the manuscript.

Reviewer #3 (Remarks to the Author):

The manuscript describes the sol-gel process of creating single Co atom active sites on silicon nanocrystals for photocatalysis. The authors provide a proposed growth mechanism in which CoSi_2 seeds the formation of Si nanodomains that contain single Co atoms. The authors also provide tests of catalytic performance, showing an EQE that surpasses other heterogeneous photocatalysts and a TON more than 10 times higher than previously reported heterogeneous photocatalysts. This is important progress in the field of silicon-based SACs and, I believe, of interest to the catalysis community. The authors have done a good job of performing complimentary experiments and analysis to support their claims, and the methods provide adequate details to reproduce their results. However, there are a few pieces of data that are necessary to more fully support their claims, especially regarding single Co atoms. With a few additional pieces of data, it is my opinion that this manuscript should be accepted for publication.

The authors use STEM to image and identify the single Co atoms on Si nanocrystals that have been etched from the SiO_2 matrix. It is difficult to see the bright atoms circled in Fig 2d. I think this figure could be improved by switching this panel with a higher resolution image, such as Sup. Fig. 4e or 4f where it's easier to see the bright atoms. HAADF imaging shows bright, single atoms that are presumed to be Co on the surface of Si nanocrystals. This is a reasonable assumption considering the atomic number difference between Si and Co, but further evidence is needed. Providing an intensity map across a bright atom and comparing it to a simulated image would help unambiguously identify the atoms as Co. The authors also show EELS data with a Co L3 edge at 779 eV. A spectrum image map of a single Si nanocrystal with the Co EELS edge coinciding with the bright atoms would be irrefutable evidence that the bright atoms are indeed Co. Why do the authors so tightly crop the EEL spectrum in Fig. 2g? They show a ~60 eV range, and based on their dispersion should have a 512 eV range of data. If there are other edges present in that range, they should be identified and explained. It is suspicious to me to show such a narrow region of the spectrum.

The XRD pattern of the 800C annealed sample alluded to in line 98-99 should be provided at least as supplementary data.

SEM imaging and EDX mapping is used to verify Co distribution. A representative EDX spectrum in addition to the elemental maps should be included in Supp. Fig. 3 and/or Supp. Fig. 7 showing Co above the noise and eliminating any suspicion that the Co mapping is an artifact. The signal in Supp. Fig. 7d is

so sparse that it could be background. While this does indicate that there is no Co clustering or aggregation, a representative spectrum showing Co above the background would prove there is Co present and strengthen the claim of well-dispersed Co atoms. Did the authors consider quantifying the EDX data? It would be interesting to see how EDX quantification compares to the Co loading.

5 nm pores are measured by Barrett-Joyner-Halenda method of N₂ absorption, shown in Supp. Fig. 8. Supp. Figs. 6-7 show no indication of pores. While the crystal in Supp. Fig. 6 shows a rough surface, 5 nm pores cannot be distinguished in a 15 μm wide crystal. If the authors believe pore size is a crucial aspect, they should consider preparing a sample for S/TEM, either by FIB liftout or finely grinding the powder, to image the pores directly. Regardless, Supp. Figs. 6-7 should not be used as a reference to pore size without further explanation.

The authors should consider revising lines 50-64 to be less chronological and more authoritative in language. The introduction should also be expanded to include specific recent examples of comparable SAC to further impress upon the reader the importance of this work.

Overall, this manuscript utilizes complimentary techniques to assess the results of a sol-gel synthesis to prepare a silicon-based single-atom photocatalyst. The combined structural analysis mostly supports their claims of a Co@Si SACs embedded in a SiO₂ matrix. Their tests of catalytic performance are very promising and exceed performance of other heterogeneous photocatalysts.

We sincerely thank all reviewers for their much-valued suggestions, which have enabled us to improve the manuscript. Following are the detailed actions taken in light of reviewers' comments:

Reviewer #1 (Remarks to the Author):

This is an exciting contribution from the authors reporting new single-atom catalysts (SACs) for solar CO₂ reduction. In this manuscript, Co@Si SACs were rationally synthesized and thoroughly characterized with a variety of techniques. The SACs demonstrated interesting activity in producing syngas using Ru-bpy as photosensitizer and TEOA as sacrificial electron donor. This work represents a novel preparation of SACs. The manuscript is publishable after addressing a few comments.

1. Figure 3b: please remove the cycles for the synthesized SACs. The cycles only served to blur the spectra. It would really help the readers to present the original spectra here and include the fitting curves in SI.

Response: We thank the reviewer for the suggestion. We have removed the fitting curves from the original Figure 3b, and we provided the corresponding fitting results in the revised Supplementary Figure 8 (see below).

Revised Fig. 3 | X-ray absorption analysis of Co@Si SACs. a, Co K-edge XANES spectra and **b,** Fitted Co K-edge EXAFS spectra of various Co@Si and control samples. The original spectra are available in Supplementary Fig. 8. **c,** The schematic shows the influence of Co concentration on the distribution of Co atoms on Si NCs. **d,** Simulated coordination number

and e, corresponding atomic distance values extracted from the Co K-edge EXAFS spectra of Co@Si and the control Co@SiO₂ samples. Negligible values of Co-O and Co-Co contribution in the bond distance are denoted using asterisks (*).

Revised Supplementary Fig. 8 | The original Co K-edge EXAFS spectra (circles) and corresponding fitting results of various Co@Si and control samples as shown in Fig. 3b.

2. Perhaps the most confusing part of this manuscript is the conditions for photocatalysis. In most cases, it is done in the presence of Ru-bpy and TEOA. However, in Supplementary Figure 14, the amount of H₂O was varied in testing. Was TEOA also used in such testing? Figure 4b could be moved to SI since it concerns testing without the use of Ru-bpy and the activity is much lower than in the presence of Ru-bpy. It would really help the readers to (i) specify conditions in figure captions, and (ii) re-organized some of the results to improve clarity.

Response: We studied the influence on the photocatalytic conditions by varying the amount of H₂O or TEOA during the tests while the other alternative was kept consistent. The results are shown in the original Supplementary Figures 14a and 14b, respectively. In the experiments shown in the original Supplementary Figure 14a (i.e., the updated Supplementary Figure 19a), we changed the amount of H₂O from 5.6 mmol to 28 mmol and kept the amount of TEOA unchanged (7.5 mmol). The H₂/CO ratio can be tuned between 1 - 2 by varying the usage of H₂O (Supplementary Figure 19a), indicating that water plays a key role in the generation of protons. In the experiments shown in the original Supplementary Figure 14b (i.e., the updated Supplementary Figure 19b), we fixed the H₂O amount to be 5.6 mmol, but changed the TEOA amount from 7.5 mmol to 22.5 mmol. The results indicate that the amount of sacrificial agent TEOA has a minor influence on the H₂/CO ratio.

We now explain more clearly on the caption of the updated Supplementary Figure 19 (i.e., the original Supplementary Figure 14) about the conditions for those control reactions and updated the Supplementary Note 2 about the impact of water and TEOA concentrations in the syngas production.

We also followed the reviewer's suggestion and moved the original Figure 4b to the Supplementary Information (the updated Supplementary Figure 17). The updated Figure 4 now includes the comparison of the EQE values and the metal loading capacity between the SACs shown this work and the reported photocatalysts (i.e., the original Supplementary Figure 12a).

Revised Fig. 4 | Performance of photocatalytic syngas production. **a**, Syngas production rates of Co@Si SACs with various Co concentrations. The results of control samples (Co@SiO₂, Co, and Co₃O₄ powders) were placed in the grey region. Negligible values of product yield are denoted using asterisks (*). **b**, TON results and production ratio on CO production with various Co concentrations. **c**, Time-dependent CO production performance of Co@Si samples with various Co concentrations. **d**, The comparison of the EQE values and the metal loading capacity between the Co@Si SACs and the reported photocatalysts (updated Ref. 15-18,20,50-64). Detailed parameters of the materials are available in the updated Supplementary Table 4.

3. Figure 4d: please use consistent significant figures for Co concentration.

Response: We have revised the manuscript and kept the significant figures of the cobalt concentration consistent throughout the manuscript and the Supplementary Information.

4. The last paragraph before "Conclusions" and Supplementary Figure 16 are not well supported by experimental results. This mechanism is likely true and could be very useful if the authors could present more results (i.e. modeling).

Response: To elucidate the photocatalytic mechanism, we carried out a series of control experiments (Figure 4a, Supplementary Figures 15, 17 and 19). We firstly performed photocatalytic reaction on the Co-doped silica (i.e., no nanocrystalline Si), and the production of both H₂ and CO was dramatically reduced compared to the result from the reactions with Co@Si SACs (Figure 4a). We therefore conclude that the nanocrystalline Si domain can promote the photocatalytic syngas production. We next investigated the photocatalytic performance of Co-free Si@SiO₂ sample (i.e., only c-Si nanodomains embedded in the silica matrix), but we only observed the generation of H₂ (Supplementary Figure 15). These results suggest that the H₂ production is mainly assigned to the c-Si nanodomains.

To further confirm the function of the Co active sites, we liberated the freestanding SiNCs from the Si/SiO₂ and Co@Si/SiO₂ samples by HF etching, respectively. We then carried out the same photocatalytic reactions on both Co-free SiNCs and Co-doped SiNCs (Supplementary Figure 17). The performance of the CO production of the Co-doped SiNCs was over 10 times higher than that of the Co-free SiNCs. We therefore concluded that the CO production is mainly attributed to the Co active sites.

These results are consistent with the picture that the generation of H₂ and CO occurs on two distinct active sites: c-Si and well-dispersed Co atoms, as proposed as the model of the syngas production promoted by Co@Si SACs (updated Supplementary Figure 21, i.e., the original Supplementary Figure 16). We conclude that the Co@Si-assisted syngas production follows a joint reaction, i.e., the reduction of the proton occurs on the surfaces of SiNCs, whereas the CO₂RR only occurs on the Co atoms. Both H₂ and CO production processes have been found from the existent Si- and Co-based photocatalysts, respectively (such as *ACS Omega* 5, 6358 (2020); *Angew. Chem. Int. Ed.*, 54, 2980 (2015); *Nat. Commun.* 10, 2840 (2019); *J. Am. Chem. Soc.* 144, 17097 (2022), etc.) Similar joint reaction mechanisms have also been proposed for other transition-metal-based heterogeneous photocatalytic systems such as Janus particles and two-dimensional heterostructures (*Angew. Chem. Int. Ed.* 59, 22246 (2020); *Angew. Chem. Int. Ed.* 61, e202204711 (2022); *Nat. Commun.* 10, 5599 (2019). *J. Mater. Chem. A* 7, 21704 (2019); *ACS Sustain. Chem. Eng.* 9, 4206 (2021)).

We now write in the revised manuscript, *"The materials characterization and the catalytic performance investigation presented above lead us to a more detailed model of the syngas production promoted by Co@Si SACs which follows a joint reaction (Supplementary Fig. 22) (updated Ref. 65)...*

...Similar joint reaction processes have also been found from other photocatalytic systems such as Janus particles and two-dimensional heterostructures (updated Ref. 67-71).

5. Please remove tracked changes in the manuscript.

Response: We have removed all "tracked changes" notes in the revised manuscript and the Supplementary Information.

Reviewer #2 (Remarks to the Author):

I am only able to comment on the microscopic and spectroscopic evidence presented in the manuscript, as the topic of catalysis falls outside my specific expertise. Thus in the following my remarks apply narrowly to the presented XRD, SEM, TEM, EELS, XPS, and EXAFS characterization of the samples.

Overall, the presented evidence appears to be supportive of the claims, but several shortcomings and unclear points need to be explained.

1. Regarding the XRD results, I do not understand the statement on lines 109-111 that the "crystalline cobalt signal is absent from the XRD pattern" due to the fact that "cobalt and silicon have similar electronegativity". Is this really about electronegativity, and if so, can the authors please elaborate?

Response: Some key components were mistakenly removed from the original discussion. We now have fixed the sentence and explain more clearly how we determined the formation of CoSi_2 nanodomains after considering the XRD results and the electronegativity values of Co and Si:

"...Since cobalt and silicon have similar electronegativity ($\chi_{\text{Co}} = 1.88$, $\chi_{\text{Si}} = 1.90$) meanwhile the crystalline cobalt signal is absent from the XRD pattern (Figure 2a), we therefore conclude that CoSi_2 nanodomains are formed through a two-step process: (1) the disproportionation of $\text{HSiO}_{1.5}$ and (2) the redox reaction from Co(II) to CoSi_2 (in which the valence of Co is close to 0) at 500°C due to its relatively lower formation energy compared to that of metallic Co (updated Ref. 43)...".

2. Regarding the XPS results, the spectral feature assigned to Co(0) seems to be much broader in the 500C sample than in the 700C sample (as shown in Figure 2c). Is this a real effect, and if so what is its cause, or is the peak deconvolution perhaps influenced by the relatively large noise?

Response: We agreed with the reviewer that the spectral feature assigned to Co(0) should be consistent in both 500°C and 700°C samples. We have refitted the XPS result and make sure the FWHM of the Co(0) is consistent throughout the all samples (see the updated Figure below).

Revised Fig. 2c | High-resolution XPS results of cobalt 2p regions. “Sat.” refers to the satellite peak of the cobalt 2p signal.

3. Regarding the EELS results, the authors should consider showing also the Si K-edge response for the nanoparticle measured in Figure 2e. Most importantly, however, it is curious that there is no attempt to capture the Co L-edge response of the individual bright atoms, as their identity has not been firmly established. Surely the authors collected EELS maps over such bright contrast? Even if the signal-to-noise is challenging, it should not be entirely out of the realm of possibility to localize the EEL signal to those atomic positions – and indeed show the full measured spectral range at those locations to rule out the presence of other impurities.

Response: We thank the reviewer for the suggestion. We agree that, in principle, it is possible to capture an EELS signal of single atoms. However, colloidal single-atom catalysts are typically quite electron-beam sensitive and thus single-atoms can delocalize upon electron beam illumination. To reduce the electron beam damage, we have used graphene TEM grids (to reduce the background) and relatively low beam current during our electron microscopy measurement. Still, possible structural changes induced by the beam may influence the reliability of the measurements when attempting to capture EELS signal or HRSTEM image of a single Co atom.

Another challenge that hampers the ability of locating single Co atoms is electron-beam induced contamination. As a mitigation, we applied activated charcoal based pre-clean treatment for the single-atom catalysts prior to TEM measurements to absorb isolated ligands and potential organic residuals (*Ultramicroscopy*, 221, 113195 (2021)). Still, a rapid growth of carbon layer under electron beam leads to a continuously increasing background signal, which cannot be fully solved.

Finally, the amount of Co atom is quite low, hence it is quite challenging to extract background (e.g., carbon) to localize the EELS signal of those single atoms and therefore correlating with atomic positions based on EELS mapping.

Therefore, given the nature of single-atom catalysts and challenges in spectroscopic technique, we confirm the purity of the catalysts in this study by providing a full range EELS spectra below (which also answered Reviewer #3's question). In combination with the HAADF-STEM data, it is very likely that the bright spots indeed corresponded to single Co atoms. Moreover, the HRSTEM indexing in Figure 2e,f of the main text and XRD results in Supplementary Information convincingly validated the crystalline Si and presence of Co.

Supplementary Fig. R1 | Full range Electron energy loss spectrum (EELS) spectra of Co@Si NCs according to the Fig. 2g.

4. Regarding the EXAFS results, two things are unclear: - How is the "Co-Si" fraction determined exactly?- How is there so much more Co-Co present in the 1.4% and 3.4% samples in Figure 3d, even though in the spectral responses shown in Figure 3a and b, it is very hard to make out such a large qualitative difference. The authors should show the peak deconvolutions underlying these analyses.

Response: We have replotted the fitted EXAFS spectra of Co@Si samples and show the peak deconvolutions of Co@Si samples at 1.4% and 3.4% Co loadings in the revised Supplementary Fig. 9 (see below), in which the Co-O, Co-Si, and Co-Co components are extracted from the fitted curves. The presence of Co-Co in these two samples is likely due to the formation of Co-CoO_x core-shell-like clusters at the Co@Si surface due to excess of Co precursor during materials growth. A size increase of Co-CoO_x cluster from 1.4% to 3.4% Co loading is evidenced by the slight increases of Co-O and Co-Co coordination numbers as indicated in the revised Supplementary Table 3 (see below).

Revised Supplementary Fig. 9 | The Co K-edge EXAFS spectra and the corresponding fitted curves together with scatter paths from EXAFS fitting of Co@Si samples at Co loadings of 1.4% and 3.4%.

Revised Supplementary Table 3 | EXAFS fitting data at the Co K-edge of various Co@Si samples

Co loading (wt%)	Scattering Path	Coordination number	R (Å)	ΔE_0 (eV)	$\sigma^2(\text{Å}^2)$
0.4	Co-Si	2.5	2.24	-6.7	0.00347
0.5	Co-Si	3.8	2.26	-7.8	0.00357
1.4	Co-O	0.4	1.81	-8.8	0.00464
	Co-Si	3.0	2.32	-4.8	0.00328
	Co-Co	0.5	2.30	-4.8	0.00328
3.4	Co-O	0.8	1.89	5.0	0.00772
	Co-Si	2.0	2.39	2.3	0.01768
	Co-Co	0.9	2.37	2.3	0.01768

5. Regarding methods, some more elaboration is needed:- Synchrotron XAS studies should mention the measurement mode, any relevant polarization, the expected energy resolution, and any further experimental parameters. Any analysis and fitting parameters should also be explicitly disclosed.

Response: We have now added the details of XAS tests in the Methods section of the revised manuscript (page 29):

“Synchrotron XAS studies. Synchrotron XAS measurements at the Si K- and Co K-edges were carried out at the Soft X-ray Micro Characterization Beamline (SXRMB, $E/\Delta E > 10^4$) of the

Canadian Light Source (CLS, Saskatoon, Saskatchewan, Canada). Non-polarized X-rays were used for excitation and fluorescence X-rays were recorded using a 7-element silicon drift detector. XAS data were processed using Athena software in Demeter (v.0.9.26). A cubic spline function was used to fit the background above the absorption edge for the XAS spectral normalization⁷⁵. The normalized EXAFS function, $\chi(E)$, in energy space was then transformed to $\chi(k)$, where k is the photoelectron wave vector. To assess the interatomic interaction of Co atoms, $\chi(k)$ was multiplied by k to amplify the EXAFS oscillations in the low- k region. Fourier transformation (from k space to R space) of the k -weighted $\chi(k)$ with a k range of 2.5 - 11 \AA^{-1} for the Co K-edge was applied to differentiate the EXAFS oscillation from different coordination shells. Co K-edge EXAFS data recorded in R space (0.6 – 3.0 \AA) was fitted using Artemis software in Demeter with the FEFF6 program⁷⁶, in which structural parameters of Co samples were calculated, including coordination number (CN), bond distance (R), inner potential shift (ΔE_0), and Debye-Waller factor (σ^2)."

Finally, the authors should openly deposit the data – and ideally the analysis code – underlying their findings at latest upon final revision of the manuscript.

Response: We are willing to share all the original data through our experiments in an open depository.

Reviewer #3 (Remarks to the Author):

The manuscript describes the sol-gel process of creating single Co atom active sites on silicon nanocrystals for photocatalysis. The authors provide a proposed growth mechanism in which CoSi₂ seeds the formation of Si nanodomains that contain single Co atoms. The authors also provide tests of catalytic performance, showing an EQE that surpasses other heterogeneous photocatalysts and a TON more than 10 times higher than previously reported heterogeneous photocatalysts. This is important progress in the field of silicon-based SACs and, I believe, of interest to the catalysis community. The authors have done a good job of performing complimentary experiments and analysis to support their claims, and the methods provide adequate details to reproduce their results. However, there are a few pieces of data that are necessary to more fully support their claims, especially regarding single Co atoms. With a few additional pieces of data, it is my opinion that this manuscript should be accepted for publication.

The authors use STEM to image and identify the single Co atoms on Si nanocrystals that have been etched from the SiO₂ matrix. It is difficult to see the bright atoms circled in Fig 2d. I think this figure could be improved by switching this panel with a higher resolution image, such as Sup. Figure 4e or 4f where it's easier to see the bright atoms. HAADF imaging shows bright, single atoms that are presumed to be Co on the surface of Si nanocrystals.

Response: We thank the reviewer for the helpful suggestion to improve the readability of the manuscript. Following the referee's suggestion, we used the Supplementary Fig. 4f (in the original submitted version) as the new Fig. 2d in the revised manuscript, in which the bright

Co atoms can be clearly discerned. Please refer to the revised Fig. 2d and revised Supplementary Fig. 6 (see below).

Revised Fig. 2d | HAADF-STEM images of Co@Si NCs extracted from the SAC solid and functionalized by octene. Co atoms are highlighted by yellow circles (scale bar: 5 nm).

Revised supplementary Fig. 6 | Additional HAADF-STEM images of ligand-functionalized Co@Si SACs. a-b, HAADF-STEM images of ligand-functionalized Co@Si NCs at different magnifications. c-d, same areas as shown in panel c and d with yellow dashed circles, denoting

the locations where the Co atoms are likely present (Scale bars: (a) 100 nm, (b) 20 nm, (c-d) 5 nm).

This is a reasonable assumption considering the atomic number difference between Si and Co, but further evidence is needed. Providing an intensity map across a bright atom and comparing it to a simulated image would help unambiguously identify the atoms as Co.

Response: We appreciate the reviewer's advice. A quantitative analysis technique "atom counting" has been established by EMAT to quantitatively analyze both positions and numbers of atoms inside a nanomaterial (*Nature*, 470, 347 (2011)). To reliably determine the number of atomic columns from atomic resolution ADF-STEM images, it is essential to know the intensity contribution from the background and atoms that will be investigated separately. As we elaborated in the reply to Reviewer #2, catalytic active atoms in single-atom catalysts are in general beam-sensitive, leading to possible atom displacement due to inevitable interactions between atoms and electron beam. Moreover, it is not trivial to estimate the contribution of background with an inhomogeneous intensity due to electron beam-induced carbon contamination, making it even more complex to achieve a reliable quantification. Taking into account the abovementioned two issues, it is therefore highly challenging to extract structural information of a beam-sensitive material using currently available quantitative analysis in a reliable manner. Nevertheless, Z-contrast HAADF-STEM imaging together with the EEL spectrum of Co is widely accepted and standard method to identify single atoms (*Catal. Sci. Technol.*, 7, 4250 (2017)). We foresee that extra efforts on quantitative imaging analysis technique is needed to tackle challenges in quantification of atomic structure of beam-sensitive materials in the future. However, this is beyond the scope of our current work.

The authors also show EELS data with a Co L3 edge at 779 eV. A spectrum image map of a single Si nanocrystal with the Co EELS edge coinciding with the bright atoms would be irrefutable evidence that the bright atoms are indeed Co.

Response: We appreciate the reviewer's suggestion. As we elaborated in the reply to Reviewer #2, to achieve an EELS mapping in which the Co mapping coincides with the bright spots as shown in STEM images, the EELS mapping experiment should be performed at sufficiently high magnifications. However, electron beam damage on the sample would be significant high at high magnification even though graphene substrate has been applied which can alleviate sputtering of atoms (to avoid beam damage) to certain extent. We attempted to obtain EELS mapping at high magnifications. However, due to the potential beam damage on the samples, it is not trivial to obtain a reliable result to achieve the goal at the current stage.

Why do the authors so tightly crop the EEL spectrum in Figure 2g? They show a ~60 eV range, and based on their dispersion should have a 512 eV range of data. If there are other edges present in that range, they should be identified and explained. It is suspicious to me to show such a narrow region of the spectrum.

Response: We apologize for the confusion. We now provide the full range data (see Response to Reviewer #2 Question#3). There are no other peaks present, which proved there is no impurities in the synthesized single-atom catalysts.

The XRD pattern of the 800C annealed sample alluded to in line 98-99 should be provided at least as supplementary data.

Response: The XRD result of the sample annealed at 800°C is now provided as the revised Supplementary Fig. 3 in the Supplementary Information.

Revised Supplementary Fig. 3 | XRD result of the Co@Si sample annealed at 800°C for 12 h.
JCDPS Reference: Si: #27-1402; Co: #15-0806.

SEM imaging and EDX mapping is used to verify Co distribution. A representative EDX spectrum in addition to the elemental maps should be included in Supp. Figure 3 and/or Supp. Figure 7 showing Co above the noise and eliminating any suspicion that the Co mapping is an artifact. The signal in Supp. Figure 7d is so sparse that it could be background. While this does indicate that there is no Co clustering or aggregation, a representative spectrum showing Co above the background would prove there is Co present and strengthen the claim of well-dispersed Co atoms. Did the authors consider quantifying the EDX data? It would be interesting to see how EDX quantification compares to the Co loading.

Response: We took the reviewer's recommendation to heart and carried out additional SEM and EDX studies on the Co@Si samples with various Co concentrations (Revised Supplementary Figure 5 and Supplementary Table 2, see below). The updated EDX elemental analysis results are consistent with those obtained from ICP-OES measurements (Supplementary Table 1). The updated EDX mapping results (Revised Supplementary Figure 21 see below) also show highly-dispersed Co atoms in the Co@SAC sample after the photocatalytic reaction.

Revised Supplementary Fig. 5 | SEM and EDX results of Co@Si SACs. **a**, SEM images and **b**, the corresponding EDX spectra of the powders of Co@Si embedded in SiO₂ with various Co concentrations highlighted on the SEM images. Inset: zoom-in EDX results showing the presence of Co signals.

Revised Supplementary Table 2 | Summary of the elemental analysis results by EDX measurements on Co@Si SACs with various Co concentrations.

Sample	Atomic % (O)	Atomic % (Si)	Atomic % (Co)
0.4 wt%Co SAC	67.188	32.477	0.335
0.5 wt%Co SAC	55.701	43.834	0.465
1.4 wt%Co SAC	62.029	36.883	1.087
3.4 wt%Co SAC	45.471	52.544	1.985

Revised Supplementary Fig. 21 | SEM image of Co@Si SAC 1.4 wt%Co in SiO₂ after 3 h irradiation and the corresponding elemental distribution of Si, O, and Co verified by EDX mapping (scale bar: 2 μ m).

5 nm pores are measured by Barrett-Joyner-Halenda method of N₂ absorption, shown in Supp. Figure 8. Supp. Figs. 6-7 show no indication of pores. While the crystal in Supp. Figure 6 shows a rough surface, 5 nm pores cannot be distinguished in a 15 μ m wide crystal. If the authors believe pore size is a crucial aspect, they should consider preparing a sample for S/TEM, either by FIB liftout or finely grinding the powder, to image the pores directly. Regardless, Supp. Figs. 6-7 should not be used as a reference to pore size without further explanation.

Response: We agree with the reviewer that the SEM images should not be used as the reference to identify the pore size. In light of the reviewer's recommendation, we carried out TEM measurements on the ground sample of Co@Si SAC in SiO₂, in which some nanoscaled pore-like topographical structures can be identified.

Revised Supplementary Fig. 11 | Bright-field TEM images of Co@Si SAC sample with 3.4wt% Co.

The authors should consider revising lines 50-64 to be less chronological and more authoritative in language. The introduction should also be expanded to include specific recent examples of comparable SAC to further impress upon the reader the importance of this work.

Response: We have revised the introduction in order to present the features and the importance of SACs and to include more examples of the SAC structures:

“...Single-atom catalysts (SACs) have received worldwide research interests toward these goals due to their tunable and well-defined elemental coordination environments (updated Ref. 4-6)

Notable synthetic progresses have been made to achieve SACs with highly-dispersed guest atoms (updated Ref. 7-9). The precise control of the ideal configuration of the catalysis, however, remains challenging. Specifically, seeking approaches to achieve efficient dispersion of the active sites on the desired substrate is essential for the development of SACs for practical catalytic applications.”

REVIEWER COMMENTS

Reviewer #1 (Remarks to the Author):

The Reviewer appreciates the authors' effort to address review comments. The manuscript in its current form is acceptable for publication.

Reviewer #2 (Remarks to the Author):

I am mostly satisfied with the responses of the authors to my queries, and think the manuscript can be published with only minor additional revisions.

The authors should mention some of the reasonable difficulties they outlined in their responses in obtaining single Co atom EELS spectra or quantitative atom intensities, either in the manuscript or its supplement, before it is published.

The peak deconvolution shown in Revised Supplementary Fig. 9 does not look entirely correct – surely the location of the Co-Co peak should be kept constant for the 1.4 wt%Co and 3.4 wt%Co samples? Further, for the 3.4wt%Co sample, the deconvolution with the Co-Co and Co-Si peaks at nearly the same location cannot be unique, and it should be possible to obtain a range of possible values by varying their relative heights. However, it seems to me the peaks should not be at the same location.

Reviewer #3 (Remarks to the Author):

This manuscript provides a method for synthesizing cobalt-on-silicon single-atom catalysts. The SACs show promise for syngas production. This is a good addition to the field, as Si offers many advantages as a SAC support, but Si-based SACs are challenging to synthesize. The work is comprehensive and utilized myriad techniques to fully characterize the SACs and their catalytic activity. With this revision and additional work, the authors have addressed my concerns, and it is my opinion that the manuscript should be accepted for publication.

Point-to-point actions in response to the Reviewers' comments
Manuscript #: NCOMMS-22-31321-T

We sincerely thank all reviewers for their much-valued comments and suggestions. Following are the detailed actions taken in light of Reviewer #2's comments:

Reviewer #2 (Remarks to the Author):

I am mostly satisfied with the responses of the authors to my queries, and think the manuscript can be published with only minor additional revisions.

The authors should mention some of the reasonable difficulties they outlined in their responses in obtaining single Co atom EELS spectra or quantitative atom intensities, either in the manuscript or its supplement, before it is published.

Response: We thank the reviewer for the suggestion. To improve the readability of our manuscript, we added a supplementary note (updated Supplementary Note 1) in the revised Supplementary Information document. Now it reads:

“Supplementary Note 1 | More details of the quantification of single Co atoms using electron microscopy techniques

Here, we discuss challenges in quantification of Co atoms using electron microscopy techniques in our study.

One can apply the quantitative analysis technique “atom counting” to quantitatively analyze both positions and numbers of atoms inside a nanomaterial (Nature, 470, 374 (2011)). Moreover, it is possible to capture an EELS signal of single atoms. However, it should be noted that colloidal single-atom catalysts are typically quite electron-beam sensitive and thus single-atoms can delocalize upon electron beam illumination. To reduce the electron beam damage, we have used graphene TEM grids (to reduce the background and potential electron beam damage) and relatively low beam current during our electron microscopy measurement. Still, possible structural changes induced by the electron beam may influence the reliability of the measurements when attempting to capture EELS signal or quantitative atom counting for single Co atom.

Another challenge that hampers the ability of locating single Co atoms is unavoidable electron-beam induced carbon contamination. As a mitigation, we applied activated charcoal based pre-clean treatment for the single-atom catalysts prior to TEM measurements to absorb isolated ligands and potential organic residuals (Ultramicroscopy, 221, 113195 (2021)). Still, a rapid growth of carbon layer under electron beam leads to a continuously increasing background signal during EELS measurement. Moreover, the carbon contamination induced inhomogeneous background intensity, which makes it more challenging to achieve a reliable quantification of positions and numbers of single Co atom inside the catalysts.

Finally, the amount of Co atom is quite low, hence it is quite challenging to extract background (e.g., carbon) to localize the EELS signal of those single atoms and therefore correlating with atomic positions based on EELS mapping.

We foresee that extra efforts on quantitative imaging analysis technique are needed to tackle challenges in quantification of atomic structure of beam-sensitive materials in the future.”

The peak deconvolution shown in Revised Supplementary Fig. 9 does not look entirely correct – surely the location of the Co-Co peak should be kept constant for the 1.4 wt%Co and 3.4 wt% Co samples? Further, for the 3.4wt%Co sample, the deconvolution with the Co-Co and Co-Si peaks at nearly the same location cannot be unique, and it should be possible to obtain a range of possible values by varying their relative heights. However, it seems to me the peaks should not be at the same location.

Response: Motivated by Reviewer #2’s comment, we sought to refine our fitting models. We realized that it is less likely to form Co-Co bond at the 1.4 wt% case, consistent with our model shown in Fig. 3c; instead, the Co-Co bond appears at the 3.4 wt% sample with its spectral feature extending to ~ 2.5 Å. Because of that, we decided to re-fit the spectra of both samples and now obtained more reasonable peak deconvolution results (revised Supplementary Fig. 9 and Supplementary Table 3, see below). Specifically, the bond distances of Co-O and Co-Si are similar in both samples, an additional Co-Co bond is shown in the 3.4 wt% sample. Concurrently, increasing the Co loading from 1.4 wt% to 3.4 wt% results in the reduction of Co-O and Co-Si coordination but an increase of Co-Co coordination due to the Co agglomeration (updated Fig. 3, see below).

We also have revised our manuscript about the EXAFS data analysis accordingly. Now it reads:

“The emergence of the Co-O was noticed in the 1.4 wt% sample and both Co-O and Co-Co appeared in the 3.4 wt% sample, indicating that the increase of Co concentration may introduce oxidation and aggregation of Co atoms (Fig. 3d)”.

Revised Fig. 3 | X-ray absorption analysis of Co@Si SACs. **a**, Co K-edge XANES spectra and **b**, Fitted Co K-edge EXAFS spectra of various Co@Si and control samples. The original spectra are available in Supplementary Fig. 8. **c**, The schematic shows the influence of Co concentration on the distribution of Co atoms on Si NCs. **d**, Simulated coordination number and **e**, corresponding atomic distance values extracted from the Co K-edge EXAFS spectra of Co@Si and the control Co@SiO₂ samples. Negligible values of Co-O and Co-Co contribution in the bond distance are denoted using asterisks (*).

Revised Supplementary Fig. 9 | The Co K-edge EXAFS spectra and the corresponding fitted curves together with scatter paths from EXAFS fitting of Co@Si samples at Co loadings of 1.4% and 3.4%. The detailed fitted results are listed in Supplementary Table 3.

Revised Supplementary Table 3 | EXAFS fitting data at the Co K-edge of various Co@Si samples.

Co loading (wt%)	Scattering Path	Coordination Number	R (Å)	ΔE_0 (eV)	$\sigma^2(\text{Å}^2)$
0.4	Co-Si	2.5	2.24	-6.7	0.00347
0.5	Co-Si	3.8	2.26	-7.8	0.00357
1.4	Co-O	1.1	1.83	-9.0	0.00474
	Co-Si	4.0	2.31	-3.6	0.00496
3.4	Co-O	0.5	1.82	-9.0	0.00577
	Co-Si	3.1	2.31	-5.1	0.01991
	Co-Co	1.4	2.37	-0.1	0.01942

REVIEWERS' COMMENTS

Reviewer #2 (Remarks to the Author):

My thanks to the referees for addressing my comments, I have no further objections for accepting the manuscript without further delay.